# Optimizing Indoor Airport Navigation with Advanced Visible Light Communication Systems [note 1]

**DOI:** 10.3390/s24165445

**Published:** 2024-08-22

**Authors:** Manuela Vieira, Manuel Augusto Vieira, Gonçalo Galvão, Paula Louro, Pedro Vieira, Alessandro Fantoni

**Affiliations:** 1DEETC-ISEL/IPL, R. Conselheiro Emídio Navarro, 1949-014 Lisbon, Portugal; mv@isel.pt (M.A.V.); a45903@alunos.isel.pt (G.G.); plouro@deetc.isel.ipl.pt (P.L.); pedro.vieira@isel.pt (P.V.);; 2UNINOVA-CTS and LASI, Quinta da Torre, Monte da Caparica, 2829-516 Caparica, Portugal; 3NOVA School of Science and Technology, Quinta da Torre, Monte da Caparica, 2829-516 Caparica, Portugal; 4Instituto de Telecomunicações, Instituto Superior Técnico, 1049-001 Lisbon, Portugal

**Keywords:** visible light communication (VLC), wayfinding, indoor navigation, user behavior, agent-based simulator, edge/fog architecture

## Abstract

This study presents a novel approach to enhancing indoor navigation in crowded multi-terminal airports using visible light communication (VLC) technology. By leveraging existing luminaires as transmission points, encoded messages are conveyed through modulated light signals to provide location-specific guidance. The objectives are to facilitate navigation, optimize routes, and improve system performance through Edge/Fog integration. The methodology includes the use of tetrachromatic LED-equipped luminaires with On–Off Keying (OOK) modulation and a mesh cellular hybrid structure. Detailed airport modeling and user analysis (pedestrians and luggage/passenger carriers) equipped with PINPIN optical sensors are conducted. A VLC-specific communication protocol with coding and decoding techniques ensures reliable data transmission, while wayfinding algorithms offer real-time guidance. The results show effective data transmission and localization, enabling self-localization, travel direction inference, and route optimization. Agent-based simulations demonstrate improved traffic control, with analyses of user halting and average speed. This approach provides reliable indoor navigation independent of GPS signals, enhancing accessibility and convenience for airport users. The integration of VLC with Edge/Fog architecture ensures efficient movement through complex airport layouts.

## 1. Introduction

The problem of difficulty in wayfinding in crowded indoor environments refers to challenges individuals face when navigating complex indoor spaces, such as shopping malls, airports, hospitals, or large office buildings, where traditional navigation methods like GPS are often ineffective or unavailable. In such environments, people may encounter obstacles such as crowded pathways, confusing layouts, or a lack of clear signage, making it challenging to find their way efficiently and effectively. This can lead to frustration, stress, and even safety concerns, especially for those unfamiliar with the environment or individuals with visual impairments. It is extremely important that passengers reach their destination fast and safely, and business travelers want to keep their waiting time to a minimum and make good use of it.

Addressing this problem is crucial to improving accessibility, enhancing user experience, and ensuring efficient movement within indoor spaces, particularly in densely populated areas where effective navigation is essential for both practical and safety reasons. Utilizing innovative technologies, such as visible light communication (VLC), can offer promising solutions to overcome these challenges and provide users with reliable guidance and direction in crowded indoor environments.

Indoor positioning methods today primarily rely on technologies such as Wi-Fi, Bluetooth, radio-frequency identification (RFID), visible light communication (VLC), and inertial navigation [1,2,3,4,5]. Despite the availability of various techniques, including Wi-Fi-based [6,7] and visual indoor topology localization [8,9], these often necessitate extensive coverage of Wi-Fi access points or expensive sensors to ensure accurate localization. VLC is an efficient data transmission technology [10,11,12] that can be seamlessly implemented in indoor environments by utilizing existing LED lighting infrastructures with minimal modifications [13,14,15].

In our approach, we propose leveraging modulated visible light emitted by low-cost white LEDs. These LEDs can switch between different light intensity levels at a rapid pace, undetectable by the human eye, allowing them to serve a dual purpose: providing illumination and enabling communication. A mobile receiver equipped with a photodetector demodulates and decodes the electrical signal generated by the modulated light, facilitating data transmission.

Furthermore, multicolored LED-based luminaires offer additional opportunities for signal modulation and detection in VLC systems [16]. White polychromatic LEDs, in particular, support wavelength division multiplexing (WDM), which significantly enhances the data transmission rate. We utilized a WDM receiver based on an amorphous SiC double PIN/PIN heterostructure acting as a light-controlled filter [17,18]. This device multiplexes various optical channels, processes them through amplification, switching, and wavelength conversion, and finally decodes the transmitted signals to recover the information.

To utilize visible light communication (VLC) for indoor navigation, research is necessary to design LED arrangements that optimize communication performance while addressing illumination constraints for various large indoor layouts. The proposed approach involves dividing the indoor space into spatial beams originating from different light sources, with each beam identified by a unique, timed sequence of light signals. The receiver, equipped with an a-SiC:H PINPIN photodiode, plays a crucial role in determining its physical position by detecting and decoding the light signals transmitted by the luminaires. The overlap of different light beams at the receiver is leveraged to increase positioning accuracy, enabling fine-grained indoor localization. By implementing this method, indoor navigation can be significantly improved, offering users precise location information and enhancing their overall experience in crowded indoor environments [19,20].

This fine-grained indoor localization capability opens up opportunities for several applications. For instance, airports are large and complex environments with multiple terminals, gates, shops, and facilities. Fine-grained indoor localization can provide passengers with precise directions to their boarding gates, check-in counters, baggage claim areas, lounges, and other amenities within the airport. This can reduce confusion, save time, and enhance overall passenger experience. Airports have a multitude of assets, such as luggage carts, maintenance equipment, and ground support vehicles. Fine-grained indoor localization can enable the real-time tracking and management of these assets, ensuring they are efficiently deployed and utilized throughout airport premises. They have stores, restaurants, duty-free shops, and other services. With indoor localization, airport retailers can offer personalized promotions, wayfinding assistance, and location-based services to passengers based on their current location within the terminal. This can improve sales, customer satisfaction, and overall revenue generation. Overall, fine-grained indoor localization in airports has the potential to streamline operations, improve passenger experience, and enhance safety and security measures, making air travel more efficient and enjoyable for everyone involved.

This paper is organized as follows: In the introduction, the challenges faced in navigating multi-terminal airports are presented, the existing literature is reviewed, and the concept of utilizing VLC for indoor navigation is justified. Additionally, the objectives of the paper and its organization are outlined. Section 2 describes the approach used for supporting wayfinding activities in multi-terminal airports using VLC technology. It explains the process of repurposing existing luminaires for transmitting encoded messages using modulated light signals. Furthermore, it discusses the communication protocol adjusted to VLC specifications and explains the coding and decoding techniques implemented to ensure reliable transmission. In Section 3, the methodology for generating the airport model is presented, and the analysis of two types of users (pedestrians and luggage/passenger carriers) is described. Additionally, it details the development of wayfinding algorithms for guiding users to their desired destinations, the process of determining optimal paths through the venue, and the analysis of how the algorithms offer turn-by-turn directions. Section 4 describes the agent-based simulator used to control traffic for both user categories and presents and discusses the results on users halting and average speeds and different geometric scenarios. Finally, in Section 5, the conclusions summarize the key findings of the study and suggest future research directions to further enhance indoor navigation systems using VLC technology.

## 2. Methodology

The primary objective is to outline the conceptual design of a VLC-based guidance system and to define a set of use cases for its application by mobile users within large buildings.

### 2.1. Background Theory on VLC

The system model is composed of two essential modules: the transmitter and the receiver, as shown in Figure 1. These modules are pivotal for the effective functioning of the VLC technology. The transmitter module plays a vital role by converting data from the sender into an intermediary form, specifically bytes. These bytes act as a transitional step before being converted into light signals by the transmitter.

To achieve both communication and building illumination, white-light tetra-chromatic sources (WLEDs) were employed, offering distinct data channels for each chip. Figure 1a illustrates the relative positions of the transmitter and receiver. Each luminaire comprises four polychromatic white-light-emitting diodes (WLEDs) arranged at the corners of a square. At each node, only one chip is modulated for data transmission. The specific wavelengths and power densities used for data transmission are as follows: red (R: 626 nm, 25 μW/cm^2^), green (G: 530 nm, 46 μW/cm^2^), blue (B: 470 nm, 60 μW/cm^2^), and violet (V: 400 nm, 150 μW/cm^2^). The intensities of the other LEDs were controlled to maintain the perception of white light. The luminous intensity of each emitter was regulated by the driving current to achieve the desired white perception. These LEDs have a wide divergence angle (2 × 60°) since they are designed for general lighting purposes, enabling the broad delivery of the VLC signal across the surrounding area.

The driving current for each emitter was independently controlled to supply the respective coding sequence and frequency. In the trichromatic LED setup, each chip can be individually switched on and off to encode the desired bit sequence.

The modulator utilizes an ON–OFF Keying (OOK) modulation scheme, leveraging the presence and absence of light to represent binary “1” and “0”, respectively. This efficient encoding method optimizes data transmission through VLC technology and allows the modulator to encode data onto the light signals. The signal is then transmitted through the optical channel. At the reception end, a VLC receiver is responsible for extracting the data from the modulated light beam.

OOK modulation stands out as a practical and efficient choice for VLC applications, providing simplicity, robustness, and compatibility with existing lighting systems. Its advantages are manifold: OOK is easy to implement and maximizes bandwidth usage efficiency, requiring minimal signal processing compared to more intricate modulation methods. It seamlessly integrates with prevalent light sources, like LEDs, used for general illumination, ensuring cost-effectiveness and straightforward deployment in VLC setups. OOK’s resilience to ambient light interference, leveraging light-pulse presence or absence, bolsters data transmission reliability across diverse indoor settings. Furthermore, OOK receivers are simpler and more economical to develop and produce than those for more complex modulation techniques. Lastly, its capability to adjust light intensity levels enables adaptive data transmission rates, dynamically optimizing performance in response to environmental fluctuations.

In the receiver module, a MUX photodetector serves as an active filter for the visible spectrum. This filter utilizes a p-i’(a-SiCH)-n/p-i(a-Si-H)-n amorphous heterostructure with low-conductivity doped layers, as shown in Figure 1b. The intrinsic layer (i’-) of a-SiC, which is 200 nm thick, has an optical gap of 2.1 eV, making it optimal for collecting blue light while allowing red light to pass through. The intrinsic a-Si i-layer, at a thickness of 1000 nm, is tailored to fully absorb green light and efficiently collect red light.

Due to these properties, the front diode (incorporating the a-SiC layer) and the back diode (incorporating the a-Si layer) together act as optical filters. The front diode confines optical carriers generated by blue- and red-wavelength photons, whereas the green-wavelength photons are absorbed across both diodes. This configuration ensures that blue and red lights are effectively filtered and detected, while green light is absorbed and detected by both diodes, enabling precise wavelength-specific signal processing. The photodetector converts these light signals into electrical signals, which are subsequently decoded to extract the transmitted information.

The optoelectronic sensor is designed with a double-pin heterostructure, where the first subcell’s intrinsic layer is composed of SiC and the second subcell’s intrinsic layer is made of Si. The layers are deposited using plasma-enhanced chemical vapor deposition (PECVD) on a transparent glass surface, with transparent contacts on both sides of the structure, allowing optical illumination in the visible (VIS), ultraviolet (UV), and near-infrared (NIR) spectra from both the front and back surfaces. The deposition conditions for the i- and i’- intrinsic layers resulted in excellent optoelectronic properties, with conductivities ranging between 10^−11^ and 10^−9^ Ω^−1^ cm^−1^ and photosensitivity exceeding 10^4^ under AM1.5 illumination (100 mWcm^−2^). To minimize lateral currents, which are critical for device operation, low doping levels were used, and methane was added during the deposition process. The doped layers, which are 20 nm thick, exhibit high resistivity (>10^7^ Ωcm) and optical gaps around 2.1 eV. Transparent contacts were deposited on both the front and back surfaces to allow light to enter and exit from both sides. The back contact defines the active area of the sensor. The front and back contacts are made of ITO, with an average transmission of approximately 80% from 425 nm to 700 nm and a resistivity of 9 × 10^−4^ Ωcm.

The working principle involves combining pulsed communication channels of red, green, blue, and violet lights (λ_R_, λ_G_, λ_B_, and λ_V_, respectively), each carrying a specific bit sequence. These channels are absorbed according to their respective wavelengths. The combined optical signal, or multiplexed signal, is then analyzed by reading the generated photocurrent under a negative applied voltage (−8V). The combined optical signal (MUX signal; received data) is analyzed by reading out the generated photocurrent under a negative applied voltage (−8V), with 390 nm background lighting, applied from the front side of the receiver. Figure 1c illustrates the spectral gain, defined as the ratio of photocurrents with and without an applied optical bias, with arrows indicating gains at analyzed input wavelengths: R, G, B, and V. The results demonstrate wavelength-dependent photocurrent variation. Front irradiation enhances long wavelength channels while attenuating short wavelength ones, effectively transforming the device into an active filter under irradiation. Gain exceeds unity for wavelengths above 500 nm, amplifying the green and red spectral ranges, while it drops below unity for shorter wavelengths, suppressing the violet and blue ranges. Notably, the signal strength increases notably with the wavelength, highlighting nonlinear characteristics crucial for MUX signal decoding at the receiver.

The obtained voltage is subsequently processed using signal conditioning techniques, such as adaptive bandpass filtering, amplification, triggering, and demultiplexing. These processes continue until the data signal is reconstructed at the data processing unit, involving steps such as digital conversion, decoding, and decision making [18,21].

The algorithms for decoding the coded information are relatively straightforward. The background acts as a selector, choosing one of the 2^n^ sublevels (where n is the number of transmitted channels) and associating each level with a unique n-bit binary code. The combination of the four channels under irradiation represents all 16 possible on/off states (2^4^). Each level is ordered by the corresponding gains in a 4-bit binary code [X_R_, X_G_, X_B_, X_V_], where X = 1 if the channel is on and X = 0 if it is off.

The functional principle relies on the adjustable penetration depths of the photons into the front and back diodes, which are linked to their absorption coefficients in the intrinsic front and back collection areas. Front irradiation is strongly absorbed at the beginning of the front diode and, due to the self-bias effect, increases the electric field at the back diode. Here, the red incoming photons are absorbed according to their penetration depths and wavelengths, resulting in increased collection. So, by assigning each output level to an n-digit binary code weighted by the optical gain of each channel, the signal can be decoded. Using this device, a maximum transmission rate capability of 100 kbps was achieved in a four-channel transmission.

To enable a receiver to gather data from multiple transmitters, it must position itself such that the coverage areas of each transmitter overlap. This creates a multiplexed (MUX) signal that serves both as a positioning system and a data transmitter. Grid sizes were carefully selected to prevent an overlap at the receiver from adjacent grid points. Figure 1a highlights nine potential overlaps (#1–#9), termed fingerprint regions, within the unit square cell. It shows also various receiver orientations (2–9 steering angles; δ) along the cardinal points and emphasizes the nine reference points per unit cell that determine the centroid of received coordinates (#1–#6), establishing it as the position reference. This approach offers a detailed resolution for localizing the mobile device within each cell. At last, the message will be output to the users. This approach indeed suggests the use of specialized WDM receivers equipped with pinpin photodiodes for optimal performance. Our study primarily focuses on the implementation of VLC technology using an MUX photodetector with light-filtering properties to detect a four-channel OOK transmission. This MUX photodetector also serves as an active filter for the visible spectrum. However, we understand the importance of utilizing widely available mobile phones for practical and user-friendly applications. While our current implementation relies on these low-cost photodetectors, we are also exploring the integration of VLC technology with mobile phone cameras, which can also detect light signals. This integration would leverage the existing hardware capabilities of smartphones, thereby eliminating the need for additional gadgets and enhancing user convenience.

### 2.2. Lighting Plan Layout, Architecture, and Geolocation

In VLC tracking, geographic coordinates are generated to guide users through unfamiliar buildings or lead them to their desired destinations. To achieve this, VLC utilizes cells for positioning and a central manager (CM) to maintain oversight and generate optimal routes [21].

Designing a geometry model for building interiors is complex due to the man-made nature of such structures, which often follow basic shapes. For this study, a square lattice topology for the luminaires was adopted for each level, simplifying comprehension and distance calculations between nodes using the x, y, and z axes. Each node represents a room, crossing, or exit, with paths linking the nodes. Figure 2a illustrates a 3D model of the building, where user positions are represented as *q* (*x*, *y*, *z*, *t*)*,* including horizontal positions (x, y) and the floor number (z). The ground floor is level 0, with levels above and below indicated by positive and negative z values, respectively. Each significant area within the airport layout is labeled, including terminals, carrier runways, gates, pedestrian pathways, and important facilities like restrooms and information desks.

The 3D model is based on the building’s footprint, collected from available luminaires, and displayed on the user’s receiver for orientation. Users can request the CM to target their destination and receive notifications of floor changes. Each unit cell is identified as *Ci*, *j*, *k,* where *i*, *j*, and *k* denote the x, y, and z positions, respectively, in the square unit cell of the top left node.

Introducing a groundbreaking concept, we unveil a mesh cellular hybrid structure that introduces a novel approach to network architecture. Figure 2b illustrates this framework at level 0. To establish secure pathways, the mesh network excels at facilitating peer-to-peer communication. This architectural design is specifically crafted to facilitate direct information exchange among devices (D2D), a crucial process in data-centric environments. At the core of this functionality is the capability of each WLED to emit a unique VLC signal, effectively serving as an identification beacon (L2D). Leveraging this beacon, the optical receiver accurately determines the user’s trajectory using a specialized positioning algorithm [22].

This calculated indoor route, represented as *q*(*x*, *y*, *z*, *δ*, *t*), contains vital spatial and temporal data, providing users with valuable insights into their movements within indoor spaces. For instance, when a user (pedestrian, P, luggage or passenger carrier, D) transitions from outdoor to indoor environments and requires guidance to navigate (P/I, D2I), they can personalize their points of interest for wayfinding services. To each user is associated by the CM an average speed that depends on the kind of device (walk, moving walkway, or passenger carrier). The requested information is then transmitted to the appropriate receiver (I2D) by emitters located within the infrastructure, such as traffic light (TL) controllers or signboards, positioned at crosswalks.

### 2.3. Communication Protocol, Coding, and Decoding Techniques

Data transmission is achieved synchronously through a 64 bit data frame structure. Information is encoded using an On–Off Keying (OOK) modulation scheme, which involves switching the signal on and off to represent binary data.

Each luminaire incorporates WLEDs (RGBV; refer to Figure 1), facilitating the simultaneous transmission of four signals and necessitating a four-channel filtering receiver to quadruple bandwidth. RGBV signals, with wavelength-calibrated amplitudes, yield 2^4^ optical combinations and 16 photocurrent levels at the photodetector, given each infrastructure’s four independent emitters. The PIN-PIN demultiplexer plays a vital role in decoding by accurately retrieving the original message based on prior knowledge of calibrated amplitudes. The communication protocol, outlined in Table 1, governs covering synchronization (Sync.), information exchange (COM), identification (q(x, y, t), direction (δ; N, S, E, W), and payload sections of the transmitted frame. The frame structure is systematic and standardized, contingent on the type of communication (COM; 1–6).

The communication protocol for the airport VLC system is structured with several key components to ensure efficient data transmission and synchronization:
*Start of Frame (SoF):* This synchronization block, consisting of 5 bits ([10101]), marks the beginning of a frame and helps synchronize all receivers with the transmitters.*Identification (ID) Blocks:* These blocks use binary representation for coded decimal numbers to encode various details, including:
○*Communication Type (COM):* Specifies the type of communication, such as location (L/I 2 D/P) or direction (D/P 2 I).○*Transmitter Localization (Position)*: Indicates the x and y coordinates of the transmitter.○*Timeline Information:* Contains time data, distinguished by the pattern END [111], including hour, minute, and second information.*Other ID Blocks:* These blocks are dependent on the communication type, and include additional identifiers required for bidirectional communication, such as:
○*Device Number (Device. Nr.):* Unique identifier for each device.○*Temporary Identification (Device. ID):* Temporary ID for communication sessions.○*Lane Occupancy (Lane 0–7):* Indicates lane usage.○*Requested Traffic Signal (TL 0–15):* Specifies the requested traffic signal.○*Cardinal Direction (N, S, E, W):* Indicates the direction of movement.○*Active Phase:* Provided by the infrastructure in “response” or “request” messages at intersections.*Traffic Message* (*Body of the Message)*: This section carries the main information, including:
○Carrier Information: Contains device IDxy (x, y coordinates), device order behind the leader (Nr. behind), and requests for intersection crossing permissions.○Traffic Payload: The main data being transmitted.*End of Frame (EoF):* A 4 bit block ([0000]) that marks the end of the transmission frame.

This structured protocol ensures the efficient encoding and decoding of critical movement information for vehicles and pedestrians within the airport VLC system, maintaining synchronization and data integrity throughout the communication process.

To decrypt the received information via the photocurrent signal as captured by the photodetector, a crucial step is undertaken. This process relies on retrieving information through a calibration curve to facilitate this mapping [23]. This curve has been pre-established taking into account the filtering properties of the PINPIN photodetector (Section 2.1).

The calibration curve is a sequence of bits carefully designed to correspond to each possible decoding level. This curve acts as a guide, helping to establish associations between photocurrent thresholds and specific bit sequences. Figure 3 presents the MUX/DEMUX signal of the calibrated cell, with a random signal (message) superimposed within the same timeframe. On the right side of the figure, the correspondence between each threshold (RGBV) and photocurrent level is illustrated, demonstrating how different photocurrent levels map to specific bit sequences. This method ensures the accurate decoding of transmitted information based on the calibrated photocurrent thresholds.

The message within the frame begins with a header labeled “Sync”, consisting of a 5 bit synchronization block [10101] applied simultaneously to all emitters. Following this, a “calibration” block is transmitted, which includes the simultaneous transmission of four calibrated R, G, B, and V optical signals. The bit sequence in this block is designed to cover all 16 possible combinations of the four RGBV input channels (Section 2.1).

After the calibration block, a random message is transmitted. To ensure accurate decoding, the periodic retransmission of the calibration curve is necessary to maintain an accurate correspondence to the output signal. The results show that the MUX calibrated signal displays distinctly separated levels corresponding to the various on/off combinations of the input channels, facilitating the decoding process.

By comparing the calibrated levels (d_0_–d_15_) with the assigned 4-digit binary [RGBV] codes for each level, decoding becomes straightforward. This comparison allows for accurate message decoding. Finally, a parity check is performed after the word has been read to ensure data integrity [24]. Upon decoding the MUX signals and considering the frame structure (Table 1), details such as pose, transmitter type, and traffic message are revealed.

## 3. Scenario

The capacity of an airport is intricately linked to its gateways, boarding areas, and the layout of aircraft door areas. The assignment of these areas can be a coordinated process, tailored to specific objectives and criteria established for each scenario. Some objectives may focus on enhancing customer service, such as minimizing the distances pedestrians or passenger/luggage carriers must travel within the airport, whether it is for landing, transit, baggage claim, terminal changes, or shopping.

### 3.1. Airport Model Generation

While there is limited knowledge specifically about pedestrians’ walking speeds within airport terminals, substantial research has been conducted on pedestrian behavior in related environments [25,26,27]. Pedestrians tend to slow down as they approach decision points along their travel path. Such situations occur at junctions, including entrances to other corridors, terminals, concession areas, and airport destinations like gate boarding areas and baggage claim areas. Empirical observations of pedestrian movements in airport terminal corridors indicate no significant difference in mean walking speeds compared to pedestrians in other transportation facilities. Similarly, observable pedestrian characteristics have little effect on walking speeds in airport corridors. However, there are instances where airport pedestrians opt to reduce their walking speeds, especially in areas incentivized by information systems to do so.

Airports are evaluated across different categories, including walking distances between terminal facilities and accessibility to ground transportation and parking. The research presented contributes to a better understanding of pedestrian behavior in terminal corridors. This understanding can inform more decisions when designing terminal facilities. In this study, the airport model generation is based on the footprints of a multi-level airport collected from available sources (RGBV luminaires: Xi,j) and displayed on the user receiver for orientation (refer to Figure 2).

Figure 4 emphasizes the main components and flow of the system, highlighting critical elements, such as VLC transmission points (Xij), user detection (the generated footprints (1–9)), and the connected luggage and pedestrian flow. The simulated scenario includes two distinct terminal intersections (Terminals 1 and 2), each featuring two four-way arms (N, S, E, and W) with two moving lanes each (L 0–7) and two sidewalks per arm. The road network topology considers terminals at intervals of 160 m, 250 m, and 400 m. Additionally, areas along the sidewalks and inside the terminals where pedestrians can shop, dine, or rest are considered.

Figure 4a presents the simulated scenario with the terminal intersections, road network topology, and pedestrian areas clearly outlined. Figure 4b provides a detailed schematic diagram of Terminal 2 with coded lanes (L/0–7) and traffic lights (TL/0–15).

The traffic management system considers four main flows of carriers along cardinal points, employing road request and response segments that offer binary choices, such as turning left/straight or turning right. Additionally, the system includes exclusive passenger lanes, waiting areas, and crosswalks. Traffic flow of luggage/passenger carriers is directed from compass directions, with lanes providing options for movement: right lanes for right turns or straight paths, and left lanes exclusively for left turns. Central traffic light systems, denoted as TL 0–15, are controlled by a central manager (CM) to regulate carrier flow and prevent collisions.

Pedestrians on sidewalks have the freedom to move in both directions. It is a prerequisite that destinations can be targeted by user requests to the CM within a specific request distance (D/P2I) and any floor changes notified if applicable. The route through the airport is communicated to the user via a response message (message distance range; I2D/P) transmitted by the traffic signals. These signals also function as routers or mesh/cellular nodes, as illustrated in Figure 2.

This request/response framework provides landmark-based instructions to help carriers identify decision points where a change in direction is necessary (action). Furthermore, it offers information to users to confirm that they are on the correct path. Figure 5a visually outlines intersection phase progressions (actions) within a structured cycle length, comprising eight carrier phases in the lanes and an exclusive pedestrian phase for the walking passengers. Each phase is subdivided into discrete time sequences, providing a comprehensive temporal framework [29].

To further enhance the understanding of the system, we included a flowchart in Figure 5b that illustrates the VLC wayfinding algorithm. This flowchart outlines the steps taken by the system to detect user positions, identify lanes, determine whether the user is a pedestrian or a vehicle, and provide optimal path instructions. In Figure 5c, the flowchart to find the optimal path is displayed.

### 3.2. VLC Wayfinding Algoritms

In this section, the algorithms developed for guiding users through indoor spaces are described step by step. Explanations of turn-by-turn directions, landmark highlighting, alerts, and alternate route suggestions are presented.

The flowchart, in Figure 5b, provides a clear step-by-step representation of how the algorithm operates, and below the steps are described in detail.

*Start:* Initialization of the system.*User Inputs Destination:* The user specifies their destination within the airport.*Detect Current Location:* The system uses VLC signals to determine the carrier and user’s current location.*Calculate Optimal Path*: The system calculates the optimal path considering:
○Four main flows of carriers along cardinal points (N, S, E, and W).○Right lanes for right turns or straight paths, left lanes exclusively for left turns.○Exclusive passenger lanes, waiting areas, and crosswalks.*Transmit Navigation Instructions (I2D/P)*: The calculated path is communicated to the user via traffic signals, including any necessary changes in floor levels if applicable.*Receive and Display Instructions:* The user’s device receives the instructions and displays them.*User Follows Path:* The user follows the displayed path, adhering to the directed lanes and traffic management system.*Check for Re-Routing:* The system continuously monitors for any changes in the environment, such as new obstacles, changes in crowd density, or traffic light signals (TL 0–15) controlled by the central manager (CM).
○If re-routing is needed (e.g., to avoid collisions or adapt to new requests/responses), the system recalculates the optimal path considering the current traffic conditions and rules.○If no re-routing is needed, the system proceeds to the next step.*Arrive at Destination*: The user arrives at the specified destination.*End:* The process concludes.

Flowchart to find the optimal path in detail.
*Position Detection*: Before calculating the optimal path, the system detects the current position of the entity (car or pedestrian).*Lane and Destination Detection/Identification:* Next, the system identifies the current lane and the intended destination.*Entity Classification*: The system then determines whether the entity is a car or a pedestrian to provide the most appropriate guidance.*Best Lane Detection*: The system checks if the entity is already in the optimal lane for reaching the destination.*Lane Change Decision:*○If Not in Best Lane: If the entity is not in the optimal lane, the system instructs it to change lanes.○If in Best Lane: If the entity is in the optimal lane, the system instructs it to continue in the current lane.*Instruction Delivery:* The system proceeds to send instructions to the entity, either to change lanes or to continue in the current lane.

Considering Figure 2 and Figure 4, along with the communication protocol outlined in Table 1 and the technique for decoding calibrated signals emitted by transmitters (Figure 3), Figure 6a illustrates the decoded optical signals (at the top of the figure) and the MUX (multiplexed) normalized photocurrent signals over a specified time frame. This diagram is shown in both a D2D (COM 2) and D2I (COM 3) communication scenarios, involving a leader device on the moving walkway at a specified position (R_3,10_, G_3,11_, B_4,10_). This device is communicating with the CM at the second terminal (T2) on lane L0 (direction E) at 10:25:46 and is followed by three other devices (nr), D_1_, D_2_, and D_3_, with the same direction, located at positions (IDx,y) R_3,8_, G_3,6_, and R_3,4_, respectively (refer to Figure 4).

In Figure 6b, the responses (I2D and I2P) from two traffic lights (TL10 and TL13) to the crossing request from the preceding carrier, *d_o_* (R_3,10_, G_3,11_,B_4,10_), and a pedestrian, *q_1_*, located in the “waiting corner” of the first terminal (R_3,4_, G_3,5_) are exemplified. The timestamps “10:25:46” and “10:28:36” represent the times at which the two responses were sent respectively for the pedestrian (COM 6) and for the passenger/luggage carrier (COM 4), and provide a reference point for when each response was generated.

### 3.3. Comparing Framework Algorithms with Empirical Data: Insights from Experimentation

Figure 7a depicts the MUX signal transmitted in a frame time to the traffic lights (TLs) by two pedestrians (COM 5) waiting at corners (P_1,2_2I) to cross the terminals (T1 and T2). The top portion illustrates the decoded messages, while the content of the message is delineated on the bottom.

In this scenario, two pedestrians at an airport aim to change terminals. Figure 7b illustrates the MUX signal sent by the traffic lights (COM 6) to both pedestrians (I2P_1,2_). The top portion of the figure shows decoded messages, while the bottom portion outlines a draft of the message content. This visual representation clarifies the communication dynamics between pedestrians at various corners and the traffic lights, particularly for crossings between T1 and T2.

The results reveal the following sequence of events:*Initial Movement:* The pedestrian begins walking on the sidewalk toward the west (W) with the intention of crossing through TL14. They wait at the designated area at positions R_3,12_-G_3,13_.*First Communication:* At precisely 10:25:44, the pedestrian, P_2_, initiates communication with the traffic light (P_2_2I). By 10:25:45, they receive a response (I2P_2_). The pedestrian patiently remains in the waiting zone until the pedestrian phase becomes active.*Missed Phase:* The information from the traffic light indicates that the current active phase is N-S (phase 1), meaning the pedestrian missed their designated phase (phase 0). Consequently, they must wait for an estimated cycle time of 3 min before being allowed to cross.*Crossing:* After waiting, the pedestrian crosses the crosswalk, reaching the next intersection in about 1 min and 50 s.*Second Waiting Zone:* Upon arrival, they wait in the designated waiting zone at positions R_3,4_-G_3,5_ until the pedestrian phase becomes active again.*Second Communication:* At 10:28:35, the pedestrian, P_1_, establishes communication with traffic light TL13 at T1 (P_1_2I). The traffic light promptly responds (I2P_1_) at 10:28:36, indicating that the currently active phase is the final one in the cycle (phase 6).

By following these steps, the pedestrian efficiently navigates between terminals, utilizing the traffic light communication system to ensure a safe and timely crossing.

The interactions between pedestrians and infrastructure highlight the benefits of pedestrian communication with traffic lights. This enables pedestrians to stay informed about the active phase and waiting times, empowering them to make timely decisions based on real-time information.

## 4. Analyzing Simulation Results: Insights and Discussion

Given the wealth of data gathered through VLC, it becomes imperative to implement a system capable of intelligently managing carrier and pedestrian traffic in real time, promptly addressing all incoming maneuvering requests. The primary challenge lies in optimizing traffic control algorithms, particularly focusing on accommodating the needs and considerations of walking passengers.

### 4.1. Analyzing Traffic Dynamics: Phase Diagram in the SUMO

To assess the effectiveness of the proposed V-VLC system in multi-terminal airports, we utilize Simulation of Urban Mobility (SUMO) [30], and agent-based simulation. SUMO is a versatile, open-source traffic simulation tool that can simulate the movement of vehicles, pedestrians, and public transport in an airport environment. It allows for the adjustment of parameters like traffic density, vehicle types, and road layouts, facilitating the implementation and testing of different traffic control algorithms. This helps to enhance traffic flow, manage intersections, and oversee pedestrian crossings.

SUMO can create scenarios mirroring real-world airport conditions, complete with multiple terminals, gates, pedestrian walkways, and traffic signals. For data analysis, SUMO provides tools to collect and analyze data, such as vehicle trajectories, travel times, congestion levels, and pedestrian movements. Its visualization capabilities allow users to observe simulated traffic flow and infrastructure layouts through a graphical interface, aiding in analysis and decision making.

Moreover, SUMO’s API supports interactions with external programs, enabling the extraction of diverse traffic flow statistics. State and phase diagrams in SUMO offer insights into traffic signal dynamics and movements within the terminals. The simulation operates on an agent-based system, where agents accumulate experience to navigate traffic situations effectively, controlling traffic lights to optimize the traffic flow in walkway lanes and sidewalks [28,31,32,33,34].

Two potential approaches to gate assignment for aircraft movement, including landings and takeoffs, are outlined: The first approach involves a low carrier flow, accommodating 1800 carriers per hour. Assumptions include a carrier velocity of 9 km/h, primarily moving from east and west directions. The primary objective is to minimize the total pedestrian walking distance inside the terminal. The second approach adopts a high carrier flow, accommodating 2300 carriers per hour at the same velocity. Here, the aim is to maximize the number of passengers served by gates. Both scenarios maintain a consistent pedestrian flow of 11,200 per hour. Within these flows, it is expected that 20% of passengers/luggage carriers will turn (remain in the same terminal), while 75% will continue straight (change terminal).

In the airport context, two traffic scenarios were evaluated. The first scenario, termed the high-traffic scenario, has a cycle duration of 120 s and is capable of dispatching 2300 carriers per hour. The second scenario, with a cycle duration of 88 s, dispatches 1800 carriers per hour.

For each terminal, pedestrian flows were simulated with 7200 pedestrians at T1 and 4000 pedestrians at T2. Pedestrians were introduced exclusively on the N and S roads in both directions, starting at various distances from the intersection. This setup aims to replicate a more realistic scenario where pedestrians originate from different points, reflecting varied starting positions and paths within the airport environment.

In this scenario, all pedestrians are introduced into SUMO at a speed of approximately 1 m/s (3 km/h), closely resembling real-world pedestrian speeds. These data are integrated into the SUMO incentive system, which aims to optimize traffic flow, reduce carrier waiting times, and maximize green-light efficiency.

Figure 8a provides an overview of the SUMO environment during a simulation. Figure 8b shows a state phasing diagram for a high-traffic scenario, incorporating both carriers (2300 vehicles per hour) and pedestrians (11,200 pedestrians per hour) over two 120 s cycles. These diagrams provide insights into the dynamic behavior of traffic light signals and the movements of carriers and pedestrians within the simulated terminals. Key observations include:*Cycle Structure*: Each cycle starts with a pedestrian phase (phase 0), allowing some pedestrians to cross. This phase turns red for pedestrians after 11 s.*Carrier Phases:* Following the pedestrian phase, phases dedicated to carriers (phases 1–8) occur, continuing until the cycle ends at 123 s. The second cycle then begins, repeating the process until 247 s, marking the end of the second cycle and the start of the third.

These diagrams highlight the effectiveness of the traffic light signal timings in managing both pedestrian and carrier movements under high-traffic conditions.

For comparison, Figure 9 presents the state phasing diagram for a low-traffic scenario. Here, there are fewer carriers (1800 vehicles per hour) but the same number of pedestrians (11,200 pedestrians per hour) over two cycles.

Key differences observed are related to a shorter cycle time of 88 s (low volume of carriers) and to the pedestrian phase duration that is extended after the first cycle to benefit the pedestrian’s movement.

These adjustments reflect the system’s responsiveness to varying traffic volumes, ensuring efficient flow and minimal delays.

### 4.2. Assessing Pedestrian Dynamics: Average Speed and Halting in SUMO

To explore pedestrian behavior in sidewalk environments, we examined two key variables: average pedestrian speed and halting frequency. The average speed provides insights into how cycle durations affect pedestrian movement, while the halting frequency analyzes stationary individuals in waiting zones, offering valuable insights into pedestrian density per square meter.

Figure 10a compares pedestrian speeds under high- and low-traffic conditions on a 160 m road. The simulation begins in phase 0, the exclusive pedestrian phase, where speeds increase gradually until approximately 11 s. As cars begin to influence the scenario, pedestrian speeds decline until around 90 s, indicating more stationary pedestrians over time. For high vehicular traffic, speeds increase sharply at the start of a new cycle, decreasing again as pedestrians navigate the waiting zones. This pattern repeats until around 120 s. Speeds stabilize at an average of 1.2 m/s as pedestrians clear the area.

Figure 10b shows the same comparison under low-pedestrian-flow conditions (7200 pedestrians per hour).

The simulation starts in phase 0 (see Figure 5), the exclusive pedestrian phase, where pedestrian speeds gradually increase until approximately 11 s. As cars start influencing the scenario, pedestrian speeds decline until around 90 s in low-traffic situations, indicating a higher proportion of stationary pedestrians over time. For the lower pedestrian flow, the average speed remains higher and stable until the end of the vehicular phases, indicating that the number of pedestrians waiting does not increase. Speeds then rise sharply, signaling the start of a new cycle, with a similar pattern observed in high-vehicular-traffic scenarios until around 120 s for both pedestrian flows. Speeds decrease again as pedestrians navigate the environment and enter waiting zones. Another speed increase marks the end of the second cycle and the beginning of the third, stabilizing at an average of 1.2 m/s as all pedestrians clear the area. Notably, in low-pedestrian-flow scenarios, pedestrian speeds drop to almost zero as all pedestrians have crossed the intersections and leave the environment [22]. In Figure 10c,d, which depict halting behavior, the graph mirrors the previous speed analysis. Up to 11 s, no pedestrians are stationary due to the active pedestrian phase. Beyond this point, the number of pedestrians in waiting zones gradually increases, with peaks occurring at different times corresponding to scenario cycle durations. The onset of the second cycle is characterized by a sudden drop in halting values in the high pedestrian flow as pedestrians in waiting zones begin to move. During this cycle, fewer people are observed in the environment, as most have crossed in the first cycle, and all have cleared by the end of the third cycle, resulting in halting values reaching zero until the simulation’s conclusion. In the low pedestrian flow, halting values remain much lower and more stable during the vehicular phases for both traffic scenarios.

These results suggest that increasing the number of cycles per hour, thereby raising pedestrian average speeds, effectively reduces pedestrian travel time. By managing traffic lights throughout the entire airport network, these insights can be used to develop effective management solutions, ultimately reducing the travel time between terminals.

### 4.3. Analyzing Intersections: Road Network Topology with Various Lengths

We proceed by varying the length of the target road. As the length increases, it is expected that pedestrians will take more time to cross from one terminal to another since their speed remains constant at 3 km/h. This increased travel time leads to a decrease in the number of pedestrians in the waiting zones.

In Figure 11a, the average speeds of pedestrians for 160 m, 250 m, and 400 m (1 × 2) topologies are compared in a high–high traffic scenario with 2300 carriers per hour and 11,200 pedestrians per hour. In Figure 11c, the pedestrian flow decreases to 7200 pedestrians per hour (high–low traffic scenario). Figure 11b shows the pedestrian densities in the “waiting corners” and on the “target road” for the three topologies in the high–high scenario, while Figure 11d presents the reduced pedestrian flow to 7200 pedestrians per hour (high–low scenario).

The results indicate that, for both scenarios, the average pedestrian speed across the three topologies is similar in the first cycle, decreasing noticeably around 120 s when the second cycle begins, revealing the initial differences. Notably, the shorter path (160 m) exhibits lower speeds due to the more crowded target road. Around 240 and 180 s, marking the end of the second cycle and the beginning of the third, respectively, a more substantial reduction in speed is observed, indicating increased pedestrians waiting in the waiting zones. This reduction is less pronounced for the 400 m path due to its greater length, causing pedestrians to take more time to reach the waiting zones and spend more time in motion.

Pedestrian density, indicative of the proximity between pedestrians, varies inversely with speed. In the first cycle, a consistent trend is observed across all topologies as the number of waiting pedestrians is simulated identically in all scenarios. As expected, in the high–low scenario, the pedestrian density is much lower, mainly in the waiting corners. However, differences emerge in the second cycle: on the 160 m path, more people are waiting, reaching the other intersection more quickly and lingering in the waiting corners longer than those on the 400 m path. Toward the end of the first cycle and the start of the second, the density of waiting pedestrians increases while the density of pedestrians in motion decreases. This seemingly counterintuitive observation stems from the fact that not all pedestrians can clear the waiting zones within the allocated 8 s phase. Consequently, some pedestrians are still waiting at the end of this phase, leading to the observed increase in density immediately after a discharge phase. Comparing pedestrian density between paths connecting both intersections reveals a lower density for the 400 m path, attributed to more space for pedestrian movement in the longer path.

For the low—low pedestrian scenario, the behavior is like the one observed in the high–low scenario. Nevertheless, the average speed is higher during the vehicular phases and the pedestrian density much lower. In the first cycle, the speeds are identical due to the generation of pedestrians. However, in the second cycle, it is observed that pedestrian speeds decrease significantly on the 160 m and 250 m roads, while on the 400 m road, there is only a slight decrease. This is because there are still pedestrians moving on the sidewalks of the intermediate roads.

In Figure 12, the halting behavior for the one-hour simulation is displayed for different vehicular and pedestrian scenarios. In Figure 12a, the high–high (2300 carriers and 11,200 pedestrians per hour) and the low–high (1800 carriers and 11200 pedestrians per hour) scenarios are compared, while in Figure 12b, the comparison is made between the high–low (2300 carriers and 7200 pedestrians per hour) and low–low (1800 carriers and 7200 pedestrians per hour) scenarios.

The results from Figure 12a show that, in the high–high scenario, during the first cycle, uniform behavior is observed across all lanes since the number of pedestrians waiting is proportional to the generation of pedestrians, which is identical in all simulations. In the second cycle, differences emerge due to the length of the lanes. On the 160 m lane, there are more people waiting because they reach the other terminal faster and stay in the waiting zone. Conversely, on the 400 m lane, there are fewer people waiting. Around 120 s, at the end of the first cycle and the beginning of the second, there is an increase in the number of people waiting as the pedestrian phase is activated and they cross the crosswalk, continuing their route. However, not all pedestrians can cross in the 8 s allotted for this phase, leaving some still waiting, which causes the increase observed in the graph (right) after a clearing phase. In the low–high scenario, with a cycle time of around 88 s, a similar behavior to the high–high scenario is observed, where the number of people waiting increases right after the clearing phase. This indicates that not all pedestrians managed to cross the crosswalks due to the high pedestrian flow.

In Figure 12b and in the high–low scenario, the number of people waiting does not increase immediately after the clearing phase as it does in the high–high scenario, but rather after some time. This happens because pedestrians move from one terminal to the other and wait in the waiting zone. With fewer people, there is no disturbance in the waiting zones, allowing pedestrians to move without issues. In the low–low scenario, there are fewer people waiting, lower densities in the waiting zones, and on the sidewalks connecting the terminals with longer lanes compared to the 160 m lane, which is a behavior that aligns with expectations.

## 5. Main Contributions and Comparison with Previous Works

### 5.1. Summary

This paper presents a novel approach to indoor navigation in crowded multi-terminal airports using visible light communication (VLC) technology. The study leverages existing luminaires as transmission points, utilizing tetrachromatic LED-equipped luminaires with the On–Off Keying (OOK) modulation within a mesh cellular hybrid structure. The primary objectives are to facilitate navigation, optimize routes, and improve system performance through the integration of Edge/Fog architecture. Agent-based simulations assess the impact on user halting and average speed, demonstrating enhanced traffic control and effective data transmission, independent of GPS signals.

### 5.2. Main Contributions

In this paper, we introduce a novel approach leveraging visible light communication (VLC) to enhance indoor navigation in multi-terminal airports. Our primary contributions include:

Real-Time Guidance: Utilizing VLC for real-time user guidance, providing turn-by-turn directions, highlighting landmarks, alerting users about crowded areas, and suggesting alternate routes.

Traffic Flow Optimization: Implementing algorithms that effectively manage pedestrian and luggage/passenger carrier traffic, reducing congestion and improving overall operational efficiency.

VLC Implementation: Developing and testing VLC algorithms tailored to indoor environments, enabling reliable communication and accurate localization through modulated light signals.

Simulation Insights: Utilizing SUMO to analyze pedestrian behavior and traffic dynamics, offering valuable insights for designing efficient navigation and traffic management strategies.

### 5.3. Overview of Previous Works

Indoor Navigation (Refs [23,28,31]): Address indoor navigation within buildings or airports, providing real-time guidance to avoid crowded areas. Target pedestrians and AGV carriers in crowded buildings or airport terminals. They use VLC with pin/pin optical sensors and LED-aided systems for indoor localization and navigation and focus on wayfinding algorithms for real-time navigation assistance.

2021 (Ref. [23]): In this paper, we focused on the development of a VLC-based navigation system for large indoor environments, such as crowded buildings. The system emphasized optimization to avoid crowded areas and employed a multi-person cooperative localization system. The key contribution was the establishment of a VLC-assisted indoor navigation protocol and the evaluation of two cellular network topologies (square and hexagonal) for positioning and communication within indoor spaces.

2022 (Ref. [31]): This work expanded on the indoor navigation concept, presenting a more refined VLC localization and navigation system tailored for multi-level buildings. The focus was on bidirectional communication between infrastructure and mobile receivers, enabling accurate route guidance and geotracking within complex indoor environments. The paper introduced improvements of alert systems for crowded regions and emphasized the real-time recalculation of optimal routes.

2024 (Ref. [28]): Our work on indoor navigation within multi-terminal airports utilized VLC for real-time user guidance, specifically addressing the challenges posed by multi-terminal airport environments. The contributions included advanced wayfinding algorithms and optimal path determination using agent-based simulations. The system was designed to enhance pedestrian and luggage carrier traffic management, leveraging VLC to provide detailed turn-by-turn directions and landmark highlights.

In conclusion, while all three references propose VLC-based solutions, their applications, user focus, technological implementations, and algorithmic approaches differ significantly.

Urban Intersections (Refs [22,29]): Focus on optimizing traffic signal performance and flow in urban settings using VLC and reinforcement learning. Primarily deal with vehicles and pedestrians at intersections and utilize VLC through streetlights, headlamps, and traffic signals for V2V and I2V communications. They emphasize reinforcement learning for traffic signal optimization and highlight reinforcement learning for optimizing traffic signal scheduling.

2024 (Ref. [22] and Ref. [29]). Both papers introduced VLC-based traffic management systems for urban intersections. While Ref. [22] focused on optimizing individual intersections through V2V and I2V interactions, Ref. [29] extended this to a multi-intersection scenario, emphasizing the use of reinforcement learning for decentralized traffic signal control across multiple intersections. The primary goal was to improve overall traffic safety and efficiency by reducing waiting times for vehicles and pedestrians.

In conclusion, while all referenced papers propose VLC-based solutions, their applications, user focus, technological implementations, and algorithmic approaches differ significantly. This showcases the versatility of VLC technology across various environments, from urban intersections to multi-terminal airports and crowded buildings.

All those references are relevant to this study as they document the progression of our research in VLC-based navigation and traffic management systems. Each paper has contributed to the development of the concepts and technologies that underpin this recent work. By building on our previous research, we have been able to address the unique challenges presented by multi-terminal airport environments, further advancing the field of indoor navigation.

## 6. Conclusions and Future Work

Utilizing VLC signals presents a promising solution for enhancing indoor navigation in multi-terminal airports. Through the integration of VLC technology with existing lighting infrastructure and traffic control algorithms, significant improvements in pedestrian and carrier traffic management are demonstrated. By optimizing traffic flow, minimizing pedestrian walking distances, and providing real-time guidance to users, VLC-based systems have the potential to greatly improve accessibility and efficiency within airport environments.

Furthermore, the study involved the generation of a two-terminal airport model based on Edge/Fog architecture, establishing a communication protocol between infrastructure and users, and developing and testing VLC algorithms. The effectiveness of these algorithms in transmitting encoded messages through modulated light signals, enabling localization and positioning, and providing reliable communication between traffic signals and different user devices is demonstrated. The implementation of VLC algorithms adjusted to VLC specifications provides real-time guidance to users, offering turn-by-turn directions, highlighting landmarks, and suggesting alternate routes, resulting in the efficient management of pedestrian and carrier traffic flow to minimize congestion and improve overall operational efficiency. SUMO provides valuable insights into pedestrian behavior and traffic dynamics in multi-terminal airports. Analyses of pedestrian speeds, halting patterns, and pedestrian density across various scenarios and road network topologies reveal the impact of traffic flow, road length, and cycle durations on pedestrian movement. Additionally, the simulations highlight the potential of SUMO as a powerful tool for modeling and analyzing complex traffic systems, offering valuable insights for designing more effective navigation and traffic management strategies in multi-terminal airports. The combination of VLC algorithms and SUMO facilitates navigation assistance, real-time route tracking, and dependable guidance for users on the move or within luggage/passenger vehicles.

While our current manuscript focuses on the implementation and benefits of VLC technology, we acknowledge the importance of comparative analysis with other methods. We plan to address this in future work by conducting comprehensive comparative experiments with other prevalent indoor navigation technologies, such as Wi-Fi, Bluetooth, and RFID-based systems. This will provide a more accurate understanding of the performance improvements achieved by using VLC technology and further validate our findings. We anticipate that VLC technology will demonstrate superior localization accuracy due to its high precision in indoor environments. Utilizing multicolored LEDs and WDM allows VLC to achieve higher data rates, facilitating faster and more reliable communication essential for real-time navigation. The combination of VLC with Edge/Fog computing architectures significantly reduces latency, ensuring timely updates and instructions for users. VLC technology supports the development of dynamic, real-time wayfinding algorithms that adapt to changing conditions, providing users with optimal navigation paths and enhancing their overall experience. VLC’s ability to operate efficiently in high-density environments makes it particularly suitable for large and crowded spaces like airports, where traditional methods may struggle with interference and congestion.

Future work could also involve the implementation of reinforcement learning (RL) techniques with multi-agent systems within airport terminals to further enhance navigation and traffic management strategies [35,36,37,38,39]. This could include developing RL algorithms that enable pedestrians, luggage carriers, and autonomous vehicles to learn and adapt their behaviors in real time based on environmental feedback. Designing RL-based navigation policies allow agents to dynamically adjust their routes and behaviors based on changing conditions in the airport environment. We must explore how RL algorithms can facilitate collaborative decision making among different entities within the terminals, such as coordinating interactions between pedestrians and autonomous vehicles to ensure safe and efficient navigation. Utilizing RL techniques to optimize the control of traffic signals in the terminals, dynamically adjusting signal timings based on real-time traffic conditions and environmental factors to minimize congestion and improve pedestrian flow is necessary. By leveraging RL with multi-agent systems, future research can advance the development of intelligent navigation and traffic management solutions tailored to the unique challenges of multi-terminal airports, ultimately enhancing passenger experience and operational efficiency.

## Figures and Tables

**Figure 1 sensors-24-05445-f001:**
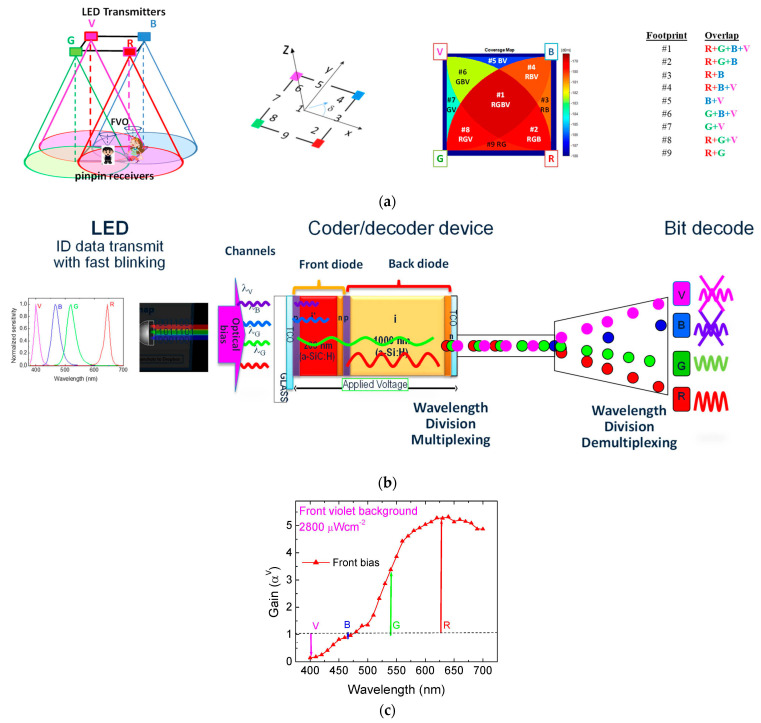
(**a**) Three-dimensional relative positions between the transmitters and the receivers, footprints, and coverage map in the square topology. (**b**) Structure and operation of the PIN/PIN receiver. (**c**) Spectral gain under violet front optical bias (α^V^). The arrows point toward the optical gain at the analyzed R, G, B, and V input channels.

**Figure 2 sensors-24-05445-f002:**
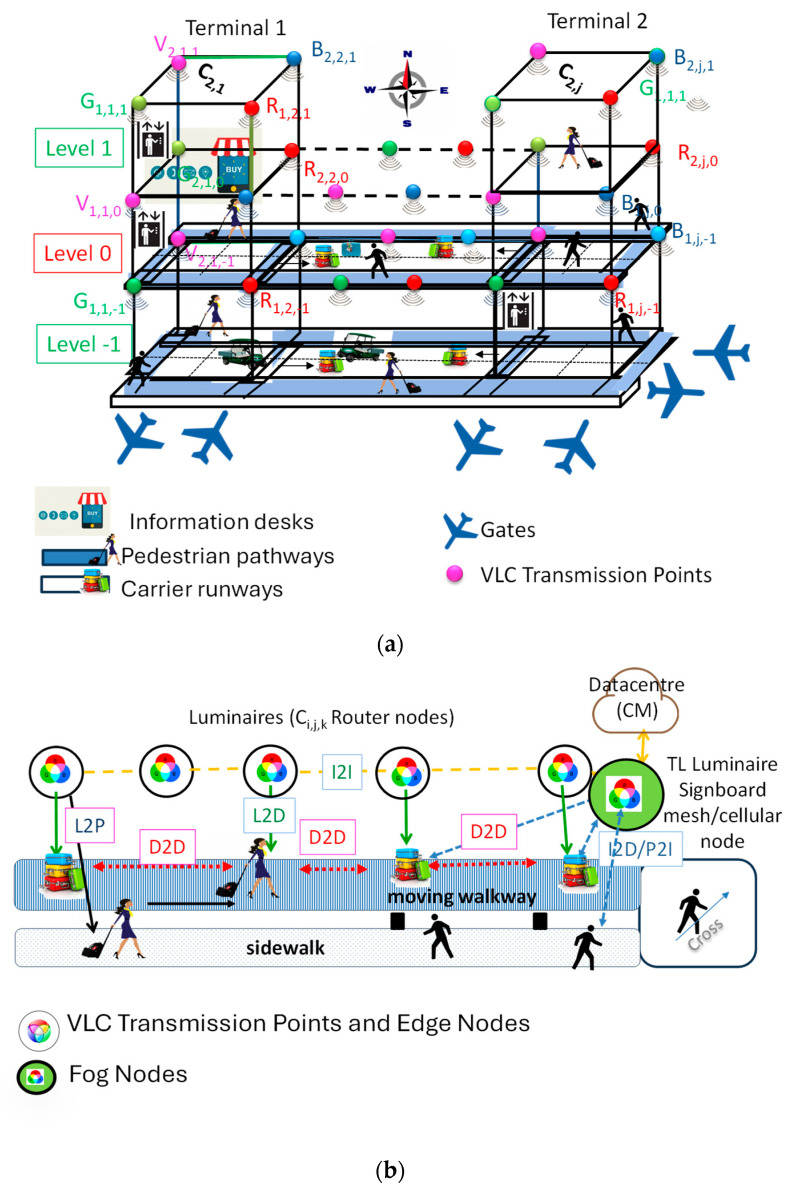
(**a**) Airport layout. (**b**) One-lane draft of the Edge/Frog hybrid architecture.

**Figure 3 sensors-24-05445-f003:**
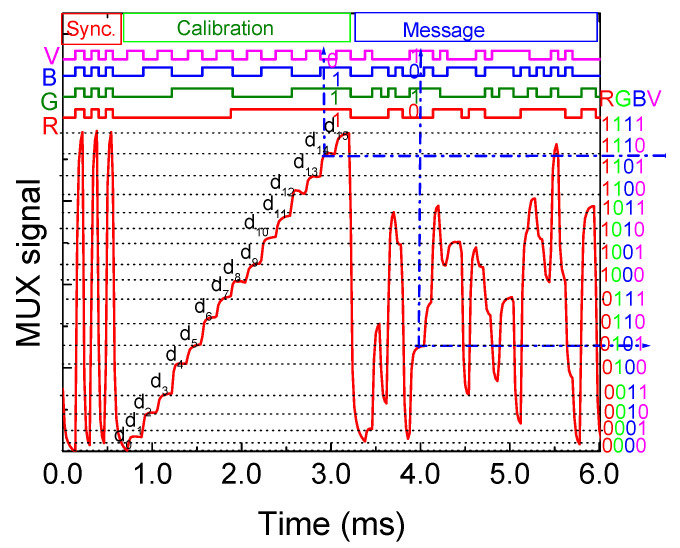
MUX/DEMUX signal of the calibrated cell. In the same frame of time, a random signal is superimposed [23].

**Figure 4 sensors-24-05445-f004:**
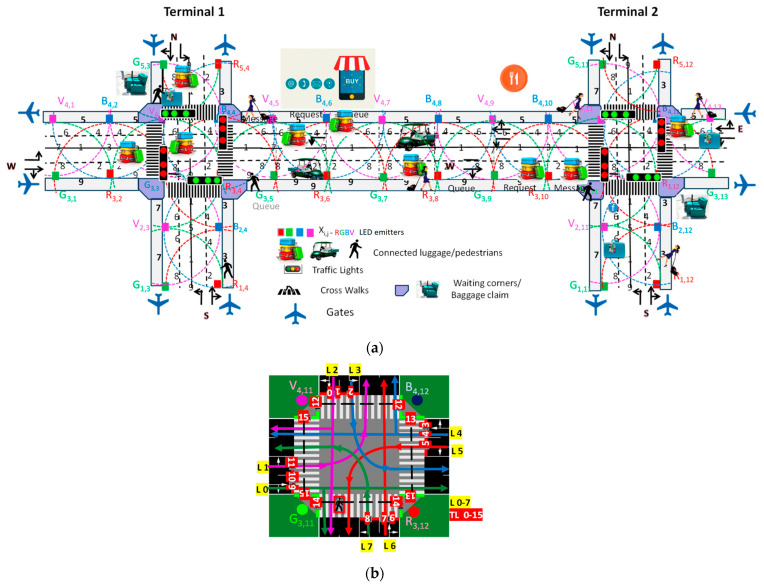
(**a**) Simulated scenario and environment with the optical infrastructure (X_ij_), the generated footprints (1–9), and the connected luggage and pedestrian flow. (**b**) Terminal 2 schematic with coded lanes (L/0–7) and traffic lights (TL/0–15). Adapted from Ref. [28].

**Figure 5 sensors-24-05445-f005:**
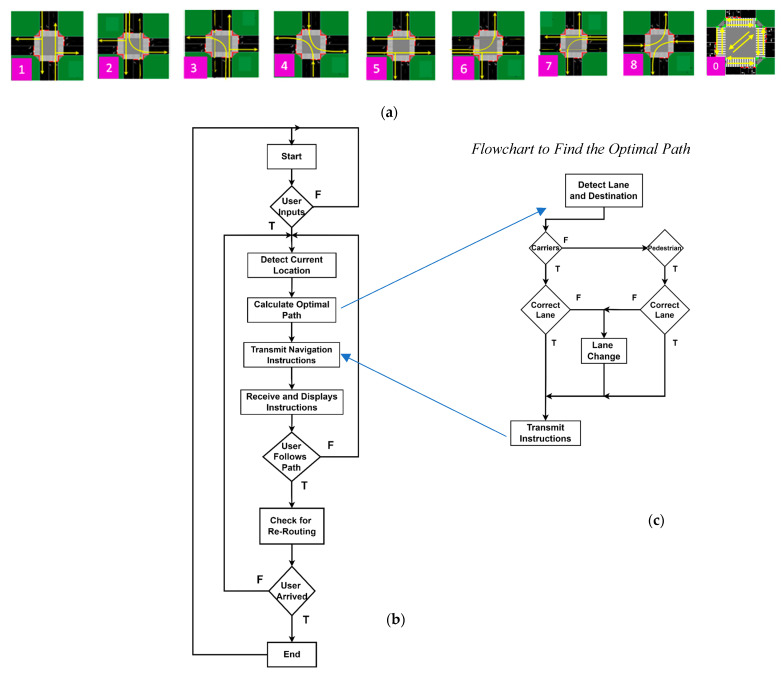
(**a**) Schematic of a one cycle phase diagram with eight moving carrier phases and an exclusive pedestrian phase. (**b**) Flowchart that illustrates the VLC wayfinding algorithm. (**c**) Flowchart to find the optimal path.

**Figure 6 sensors-24-05445-f006:**
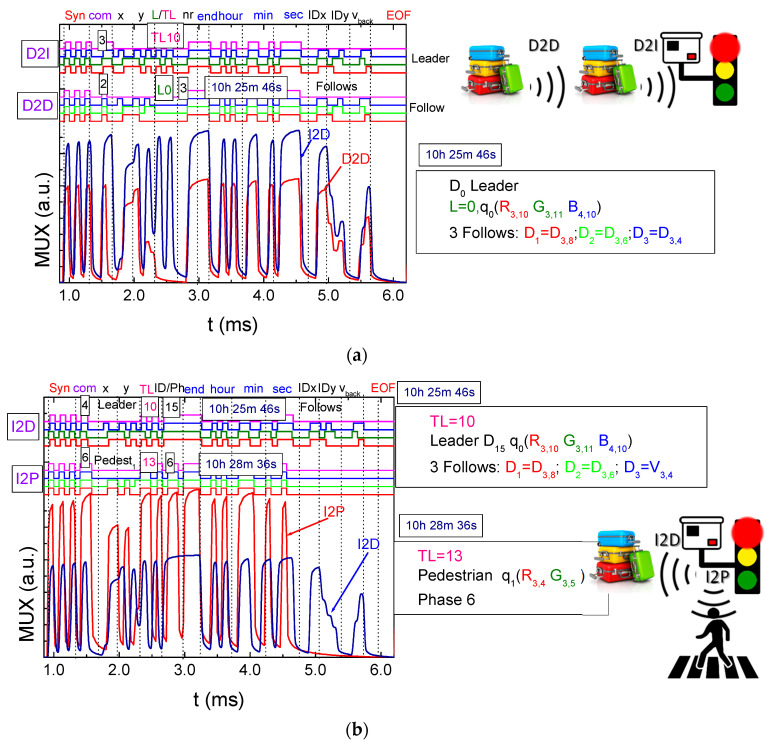
MUX signal request (**a**) and response (**b**) allocated to different V-VLC types. On the top, the decoded messages are displayed.

**Figure 7 sensors-24-05445-f007:**
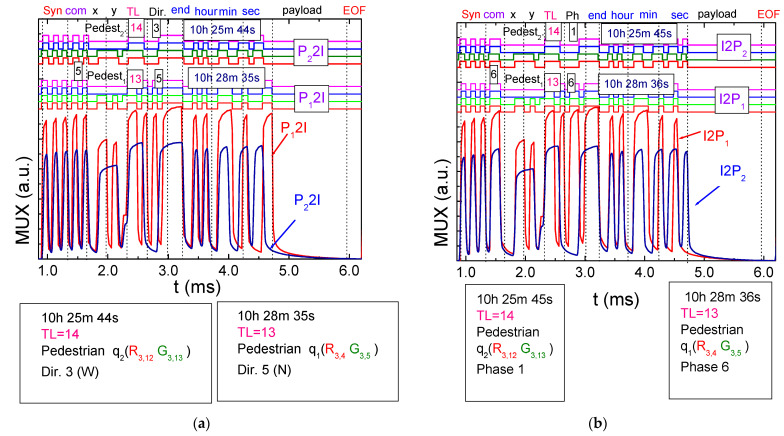
Normalized MUX signal responses and the corresponding decoded messages. (**a**) Messages sent (P_1_,_2_ 2 I) by pedestrians waiting in the corners and (**b**) messages received by them (I 2 P_1,2_) at various frame times.

**Figure 8 sensors-24-05445-f008:**
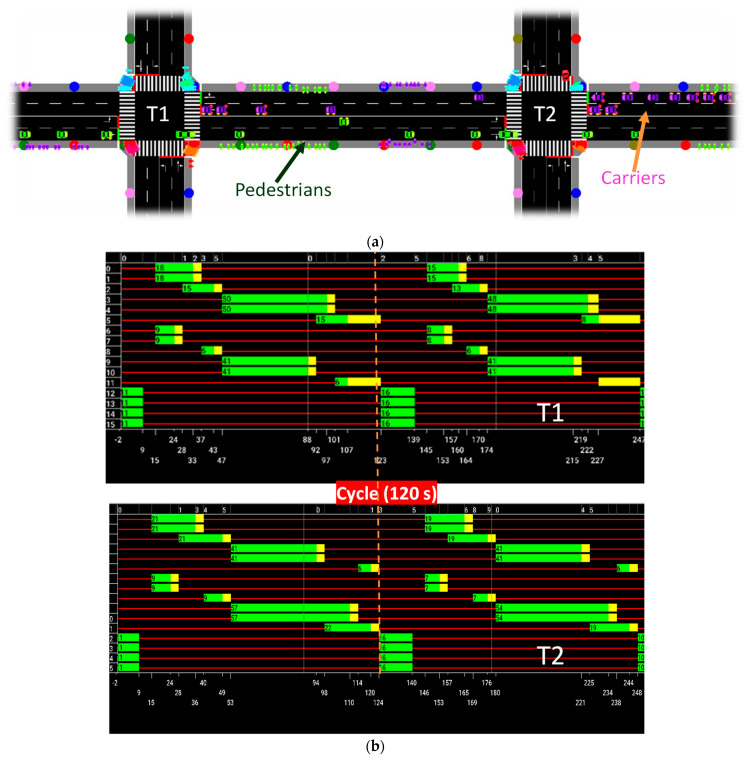
(**a**) SUMO environment. (**b**) State phasing diagram for the high-traffic scenario.

**Figure 9 sensors-24-05445-f009:**
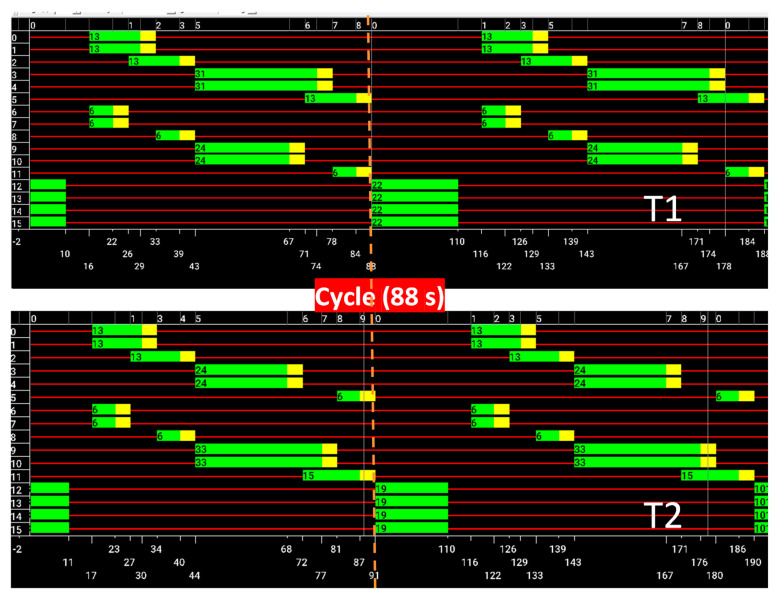
State phasing diagram for the low-traffic scenario.

**Figure 10 sensors-24-05445-f010:**
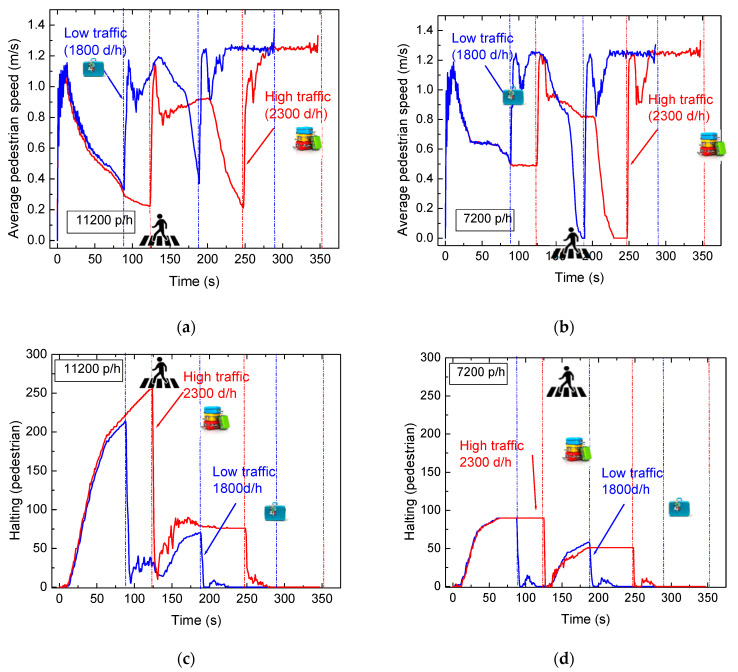
High and low vehicular traffic scenario comparison as a function of the cycle’s duration. (**a**). Average pedestrian speed (11,200 p/h). (**b**) Average pedestrian speed (7200 p/h). (**c**) Halting (11,200 p/h). (**d**) Halting (7200 p/h).

**Figure 11 sensors-24-05445-f011:**
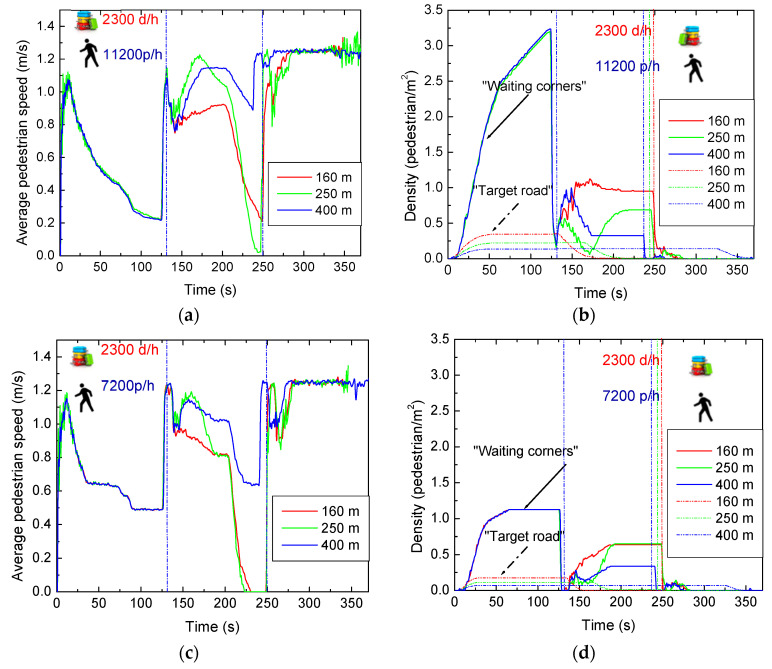
High–high scenario (2300 carriers and 11,200 pedestrians per hour): (**a**) average speed of pedestrians and (**b**) density of pedestrians as a function of time in the target road and in the waiting corners. High–low scenario (2300 carriers and 7200 pedestrians per hour): (**c**) average speed of pedestrians and (**d**) density of pedestrians as a function of time in the target road and in the waiting corners.

**Figure 12 sensors-24-05445-f012:**
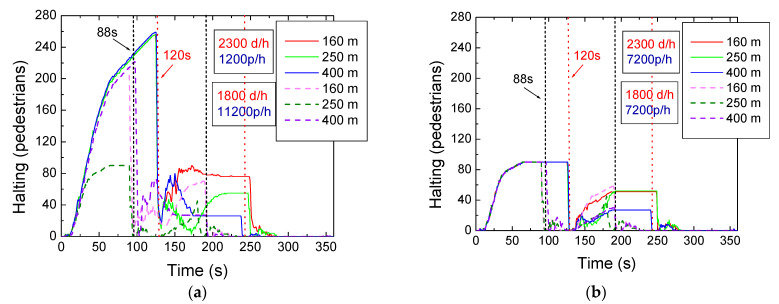
High- and low-vehicular-traffic scenario comparison as a function of the cycle’s duration. (**a**) Halting (11,200 p/h). (**b**) Halting (7200 p/h).

**Table 1 sensors-24-05445-t001:** Communication protocol: synchronization (Sync), type of communication (COM), geolocation (position and time), and payload sections of the transmitted frame.

	SOF	COM	Position		Other	ID	Timeline				Traffic Message			
L2D	Sync	1	x	y	0 bits		END	Hour	Min	Sec				EOF
D2D	Sync	2	x	y	Lane (0–7)	Device (nr)	END	Hour	Min	Sec	Device IDx	Device IDy	Nr. behind	EOF
D2I	Sync	3	x	y	TL (0–15)	Device (nr).	END	Hour	Min	Sec	Device IDx	Device IDy	Nr. behind	EOF
I2D	Sync	4	x	y	TL (0–15)	Device ID	END	Hour	Min	Sec	Device IDx	Device IDy	Nr. behind	EOF
P2I	Sync	5	x	y	TL (0–15)	N,S,E,W.	END	Hour	Min	Sec				EOF
I2P	Sync	6	x	y	TL (0–15)	Phase	END	Hour	Min	Sec				EOF

## Data Availability

No new data was created.

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
