# Peer review of "Optimizing Indoor Airport Navigation with Advanced Visible Light Communication Systems†"

_sensors, 2024, doi:10.3390/s24165445_

Round 1

Reviewer 1 Report

Comments and Suggestions for Authors

This study introduces an approach to enhance indoor navigation within crowded 18 multi-terminal airports using Visible Light Communication (VLC) technology. The work is well structure and the conclusions are good enough to be published. The authors should explain better the interaction between the user and the system. In the paper is explained the use of photodiodes to detect the VLC OOK transmission but it seems the user will requets a special gadget and not the mobile phone wich is proposed in other papers. The authors could explain this point.

Reviewer 2 Report

Comments and Suggestions for Authors

The authors provide a new approach to enhance indoor navigation within multi-terminal airports using Visible Light Communication (VLC) technology. The following are the reviewer's comments:

1- The contributions of this paper are not clear. The authors need to list their novel contributions and provide a clear comparison between the work of this submission and their previous works mentioned in [21]-[24], [27] and [28], [30] and [31].

2- In Fig.1b, some of the drawings (waveform) overlap with the writing.

3- Do the users need to carry optical receivers with them? If yes, how complex is that? Is it cost-effective?

4- On P. 9, Line 379, is parity check sufficient for data integrity? It has a well-known limitation of not identifying an even number of errors.

5- The source of information for the paragraph starting at Line 402 in P. 10 needs to be mentioned.

6- A flowchart is needed to illustrate the VLC way-finding algorithm.

7- The presented scenario in P. 13, Line 490, is very hard to follow.

8- SUMO is a known simulator. There is no need to write all these lines about it.

9- The descriptions of Fig. 8 and Fig. 9 need significant work. Also, the subsequent figures with red and blue curves need a much better explanation and presentation. The reviewer could not follow the comparison the authors made or how this highlights the contributions of this work. 

Comments on the Quality of English Language

The spelling of some words needs to be corrected, such as "waifinding". Also, the explanation of a significant number of figures is not clear.

Reviewer 3 Report

Comments and Suggestions for Authors

In this paper, the authors propose a new approach to enhance indoor navigation within crowded multi-terminal airports using VLC technology. The study employs a four-color LED illumination device and an amorphous SiC double PIN-PIN heterostructure photodetector for user positioning and path optimization, which improves accessibility and convenience for airport users in crowded and unfamiliar environments. However, there are some questions worth explaining in this paper:

1. The description in the abstract section is not sufficiently concise.

2. While the manuscript introduces an innovative approach to optimizing indoor airport navigation using Visible Light Communication (VLC), it should further clarify the unique contributions of this study and how it differs from existing technologies. For instance, adding a detailed comparison with existing research in the introduction can highlight the innovations and improvements of this study.

3. Although the manuscript presents simulation and experimental results, the analysis and discussion section could be more in-depth. It is recommended to include statistical analysis of the results, further explain the impact of different parameters on the experimental outcomes, and discuss the implications of these results for practical applications.

4. Did the manuscript consider the impact of factors like lighting conditions, obstacles, and user behavior on the system?

5. The manuscript mentions using multicolored LEDs and Wavelength Division Multiplexing (WDM) to enhance data transmission rates and localization accuracy. What were the specific transmission rates and localization accuracies achieved in the implementation? Are these metrics sufficient for practical applications?

6. Estimating the deployment and maintenance costs in large airports will be beneficial for the application of this system.

7. The figure lacks detailed labels and unit explanations. For example, parts (a) and (b) of Figure 2 do not sufficiently describe the elements within the charts.

8. In Figure 6, the signal request and response diagrams lack detailed descriptions of the time axis and signal strength units.

9. Figure 4 fails to focus on the core and the distribution of information appears to be redundant, making it laborious for the readers to capture key information quickly.

10. In Section 3, Airport Model Generation, there is a duplication of text between lines 384-389 and 391-396. In Section 4, the sentences in lines 526 and 527 are repeated.

11. There is a spelling error in the title of subsection 3.2, which should be 'wayfinding' instead of 'waifinding'.

12. It is recommended that pseudo-code be added to the 3.2 subsection to elaborate the flow of the algorithm and enhance its comprehensibility.

13. The simulation results analysis section lacks comparative experiments with other methods to give readers a more accurate understanding of the performance improvements achieved by using VLC technology.

Comments on the Quality of English Language

See the above comments

Round 2

Reviewer 2 Report

Comments and Suggestions for Authors

- In their response, the authors compared this submission with some of their relevant previous works, but not all of them. For example, in the revised manuscript, they wrote "Vieira et al. (2024)" without proper citation to which publication they meant. It is not clear which reference they mean [22] or [28]. In both cases, they should be more elaborate about each specific reference and compare this submission with reference [31], also published in 2024, which also targets multi-terminal airports.

- Moreover, they have another relevant paper published in 2021, which is not mentioned in their response (ref. [23]).  

In their response to the paper published in 2022, the authors mentioned, "Our current work builds on this by introducing an airport-specific navigation system, addressing the higher complexity and dynamic nature of airport environments." The authors need to clarify more about what is involved in the higher complexity compared with the other work and what they mean by dynamic nature. Their works in 2022, 2023, 2024, and 2021 combined appear quite similar to this work. 

Reviewer 3 Report

Comments and Suggestions for Authors

This reviewer has no further comments and thanks the authors’ revision.

Comments on the Quality of English Language

Nona

Author Response

Dear Reviewer,

Thank you very much for your positive feedback and for acknowledging our revisions. We greatly appreciate your time and effort in reviewing our manuscript.

Best regards,

Manuela Vieira